# Describing the Eye Health of Newly Arrived Refugees in Adelaide, South Australia

**DOI:** 10.3390/ijerph21070869

**Published:** 2024-07-02

**Authors:** Kate Murton, Antonietta Maldari, Joanne Thomas, Jan Williams, Marcel Nejatian, Hessom Razavi, Lillian Mwanri

**Affiliations:** 1Refugee Health Service, SA Health, Adelaide, SA 5000, Australia; antonietta.maldari@sa.gov.au (A.M.);; 2Lions Eye Institute, Perth, WA 6009, Australia; 3Centre for Ophthalmology and Visual Science, The University of Western Australia, Perth, WA 6009, Australia; 4Research Centre for Public Health, Equity and Human Flourishing (PHEHF), Torrens University, Adelaide, SA 5000, Australia; lillian.mwanri@torrens.edu.au

**Keywords:** refugee, visual impairment, refractive error, eye health

## Abstract

This study describes the eye health of newly arrived refugees attending a state-funded health service in Adelaide, South Australia, helping to address the paucity of data on the eye health of refugees. Patients attending the Refugee Health Service undergo comprehensive assessment by an on-site optometrist with accredited interpreters if they have eye symptoms, personal or family history of eye disease, or visual impairment (using World Health Organization definitions). A retrospective audit of this service was performed to obtain patient demographics, presenting best-corrected distance visual acuity (better-seeing eye), diagnoses, and management. In 2017–2018, 494 of the 1400 refugees attending the service underwent an optometry assessment (age range 1–86 years, mean age 33.1 ± 18.6 years, 53% female). Regions of origin included the Middle East (25%), Bhutan (24%), Afghanistan (22%), Myanmar (15%), and Africa (14%). Of the 124 cases of visual impairment, 78% resolved with corrective lenses and 11% were due to cataracts. Ophthalmology follow-up was required for 56 (11%) patients, mostly for cataracts (22 patients). Newly arrived refugees have high rates of visual impairment from refractive error and cataracts. Integration of optometry and state-based refugee health services may improve the timely detection and treatment of these conditions.

## 1. Introduction

Globalisation has enabled human mobility to increase, with over 300 million people living outside their country of origin for extended periods of time [1]. Among this population, at least 108 million are forcibly displaced; 35.3 million are refugees and 62 million are internally displaced due to conflict [2]. In Australia, migrants, including those with cultural and linguistic diversity, are a significant component of the population (30%), with 7.6 million people living in Australia having been born overseas [3]. Newly arrived refugees often come from a background of trauma and displacement, with multiple health issues including chronic diseases, mental health issues, nutritional deficiencies, infectious diseases, and hearing and vision loss [4].

For refugees, vision loss can be a significant barrier to learning English given the reliance on classroom board and reading exercises. This, compounded with pre-migration experiences, can impact on settlement, education, driving, and finding employment [4,5,6].

Vision loss is a major public health concern, with the World Health Organisation estimating 1.3 billion people live with some form of vision impairment or blindness [7]. An estimated 80% of vision impairment is preventable or treatable, much of it with prescription glasses [7,8,9]. The major causes of vision loss worldwide include uncorrected refractive error, cataracts, age-related macular degeneration, and glaucoma [7]. Vision loss can limit social participation, increase falls, depression, hip fractures, admission to aged care facilities, and the use of health services and has substantial economic costs to society [4,5].

In 2021 in Australia, vision loss was estimated to cost AUD $27.6 billion annually, or $46,950 per person aged over 40 [10]. This included hospital costs, productivity losses, carer costs, and the cost of aids.

A recent systematic review by Bal et al. suggested that refugees experience a high burden of eye disease, with limited access to appropriate care [4]. Five studies included in this review specifically analysed vision loss, reporting that the rates of blindness (presenting visual acuity (VA) < 3/60) within refugee camps in Africa and Asia range from 4.4% to 26.2% in clinic-based studies, and from 1.3% to 2.1% in population-based screening studies [11,12,13,14,15]. In comparison, just 0.21% and 0.31% of non-Indigenous and Indigenous people in Australia, respectively, are blind (VA < 6/60) [16]. Data from Canada support these findings, with rates of vision loss among newly arrived Syrian refugees being up to 32 times higher than the general Canadian population [17,18]. The limited available evidence suggests that the main causes of vision loss among refugees are cataracts, uncorrected refractive error, amblyopia, and corneal opacities [13,14,15].

Australia resettles thousands of newly arrived refugees every year [19]. The only published Australian study on refugee eye health reported a high rate of bilateral vision loss (27.2%) among those attending community optometrists in Victoria [20].

Given the limited data regarding eye health of refugees in Australia, the purpose of this paper is to describe the vision status and ocular morbidity of newly arrived refugees who were assessed by an optometrist in 2017–2018 at the Refugee Health Service (RHS) in Adelaide, South Australia.

## 2. Materials and Methods

The RHS is the state-funded multi-disciplinary specialist service providing on-arrival health care for humanitarian refugees and asylum seekers in South Australia. All RHS clients have a comprehensive health assessment based on the Australian Society of Infectious Diseases/Refugee Health Network of Australia recommendations for people from refugee-like backgrounds [8]. The health assessment is initiated in a nurse-led clinic, with medical follow up after two weeks. People were referred to optometry if they self-reported eye issues, had a family history of vision problems, or, in some cases, following medical assessment and testing of visual acuity.

At the time of the study, interpreters were unavailable for private optometry services in Australia. RHS was one of the only refugee health services in Australia providing optometry examination with onsite accredited interpreters funded by the service.

This study audited two years of retrospective data collected from the medical records of clients attending optometry at RHS between 2017 and 2018. Ethics approval was obtained from the Central Adelaide Local Health Network research ethics committee (Reference HREC/18/CALHN/633).

Eye health services were provided by a visiting optometrist who has experience in assessing adults and children, with an accredited interpreter in attendance. A history was taken and presenting eye symptom(s) were documented. Presenting VA was tested using a Snellen letter chart or tumbling E chart, depending on client literacy. Refractive error was measured using an auto-refractor, with subsequent subjective refinement to provide best-corrected visual acuity (BCVA). Anterior and posterior segments were examined using a combination of slit lamp, direct, and binocular indirect ophthalmoscopy. Cataracts were graded using the LOCS III scale [21]. Pupil dilation was performed with drops in people with diabetes or where otherwise indicated.

Following diagnosis by the optometrist, glasses were ordered through a brokerage with the Essilor Vision Foundation, with the majority provided at no cost to the client. Specialist and hospital referrals were made where indicated.

Demographic and clinical data were collated in Microsoft Excel and each client was allocated a unique identifier. Clients were categorised into ten-year age groups and five ethnicity groups (Appendix A). Vision loss was defined as any degree of vision impairment or blindness. The level of vision impairment (both presenting and best-corrected) was categorised using ICD-11 (2018) [22] definitions based on vision in the better eye, as: no vision impairment (VA ≥ 6/12); mild (VA < 6/12); moderate (VA < 6/18); severe (VA < 6/60); and blindness (VA < 3/60 or < 6/120) (Appendix A). Uncorrected refractive error was the primary cause of vision loss when VA improved to ≥6/12 after refractive correction. When multiple pathologies were present, the most clinically significant was chosen as the primary cause.

Data were checked for completeness and accuracy and analysed using Statistical Package for Social Sciences (version 24, SPSS Inc., Chicago, IL, USA). Descriptive statistics were analysed using chi-squared test for association. Logistical regression was performed to examine the effect of independent variables on vision impairment. Lastly, multivariate regression was used to examine the effect of covariates (sex, ethnicity, and age). The level of significance was set at 0.05. Privacy was ensured through de-identified data and secure data storage.

## 3. Results

In 2017 and 2018, 494 clients had an optometry assessment at RHS. The mean age of clients attending optometry was 33.1 ± 18.6 years (95% CI: 31.5–34.8; range of 1 to 86 years), and 52.8% were female (Table 1).

Presenting and best-corrected VA were assessed and documented in 460 (93.1%) and 470 clients (95.1%), respectively (Table 2). There were 124 clients with vision loss at presentation (27.0%; 95% CI: 23.1–31.2), which reduced to 26 clients (5.5%; 95% CI: 3.8–8.0) after refractive correction. Vision loss was comparable between females and males (27.8% and 26.0%, respectively), but significantly higher among females after refractive correction (7.9% and 2.8%, respectively; adjusted odds ratio (OR), 3.0; 95% CI: 1.2–7.7; *p* = 0.02). Vision loss increased with age, being 2.5 times more likely in those ≥50 years compared to those <50 (crude OR 2.5; 95% CI: 1.5–4.1). When adjusted for age, there was no significant correlation between ethnicity and the presence of vision impairment (*p* = 0.063).

Of the 494 clients referred to the optometrist, 433 (87.7%) had at least one ocular diagnosis, with the most common being refractive error (*n* = 328; 66.4%; Table 3). Refugees from the Middle East were twice as likely to have refractive error compared to all other ethnicities (adjusted OR 2.1; 95% CI: 1.3–3.4; *p* = 0.04). Of those with vision loss (*n* = 124), 117 (94.4%) were avoidable, with the most common cause being refractive error (*n* = 98; 81.5%) followed by cataracts (*n* = 14; 11.3%).

Corrective glasses were prescribed to 303 clients (92.3% of those with refractive error, and 61.3% of the entire cohort). Of those with refractive error, 109 (22.1%) were prescribed single vision distance glasses, 47 (9.5%) were prescribed single vision near glasses, and 147 (29.8%) were prescribed both single vision near and distance or bifocal/multifocal glasses.

Cataracts were found in 6.7% (*n* = 30) of all clients and in 17.7% (*n* = 22) of clients with vision loss. Bhutanese/Nepalese refugees were 4.8 times more likely to have cataracts compared to all other ethnicities (crude OR 5.4; 95% CI: 2.5–11.6; *p* = 0.001; adjusted OR 4.8; 95% CI: 2.0–11.4; *p* < 0.0001).

Dry eyes were found in 38.3% (*n* = 189) of all clients. After adjusting for age and ethnicity, females were twice as likely to have dry eyes (adjusted OR 1.9; 95% CI: 1.3–2.8; *p* = 0.002). Burmese and Bhutanese refugees had the highest prevalence of dry eyes with 49.3% and 47.1%, respectively. When adjusted for sex and ethnicity, clients aged 30–39 were most likely to have dry eyes (adjusted OR 2.7; 95% CI: 1.3–5.5, *p* = 0.06 against age group 0–9 years) and dry eye risk was higher in those aged less than 50 compared to 50 or older (adjusted OR 1.7, 95% CI: 1.0–2.8; *p* = 0.04).

The most common presenting symptom was ‘blurred vision’ (*n* = 200, 40.5%; 95% CI: 36.3–44.9; Table 4). There were 13 clients (2.6%) who presented with no symptoms but who were diagnosed with vision loss, and 30% of clients presented with more than one symptom.

There were 56 (11.3%) referrals made to ophthalmologists. Seven were for further assessment to aid in a diagnosis, 22 for cataracts, five for glaucoma, and four for macular degeneration. The remainder were for other diagnoses including dermatochalasis (*n* = 5), diabetic retinopathy (*n* = 3), and retinal tuberculosis (*n* = 1).

## 4. Discussion

This study provides the second ever published description of eye health among newly arrived refugees attending an Australian health service. A substantial proportion of refugees had vision loss, including some who did not report any symptoms. Much of this vision loss was treatable with glasses. These findings highlight the importance of universal eye screening and refractive services for this population, as recommended by the Australian Society of Infectious Diseases, the Refugee Health Network of Australia, and the Royal Australian and New Zealand College of Ophthalmologists [8,23]. However, appropriate resourcing and funding are essential to provide these services.

Vision is important across all stages of life. Vision enables child development, education, community participation, and participation in the workforce [24]. For people of refugee-like background, the barriers to integrating into a new country are already significant and are made more challenging with vision loss [4]. Ensuring refugees have vision loss diagnosed and corrected quickly can be expected to significantly aid with learning English, with settlement, and with seeking employment.

Of the 1400 clients who had newly arrived to RHS in 2017–2018, 494 (35.3%) were assessed by the optometrist. This sample was representative in sex but not in ethnicity or age. Clients from the Middle East were most likely to access optometry services (55%) but showed lower rates of ocular morbidity compared to clients from Africa (22.3%), Burma (27.4%), Afghanistan (32.7%), and Bhutan/Nepal (51.5%).

As expected, older clients were overrepresented, with 79.4% (*n* = 201/253) aged 40 or older, compared to 25.5% (*n* = 293/1147) aged under 40 years. While there was no difference in presenting visual acuity between females and males, females had more vision loss after refractive correction (7.9% and 2.8%, respectively; adjusted OR 3.0; 95% CI: 1.2–7.7; *p* = 0.02), indicating a higher prevalence of non-refractive ocular pathology among females, which is similar to global burden of disease studies [25].

The overall prevalence of vision loss was 27.0% (*n* = 124), which was similar to rates in a comparable Australian clinic-based study [20]. However, our prevalence of moderate-severe visual impairment (12.8%) was higher than those from a systematic review of refugee eye health (6.9%) [4]. Bilateral VA < 6/60 (Australian definition of blindness) was 1.5% (*n* = 7), which is substantially higher than the reported figures of 0.21% (non-Indigenous) and 0.31% (Indigenous) in the general Australian population [16].

There was substantially more vision loss among clients aged 50 years or over, likely due to the higher prevalence of ocular pathology among older age groups [26]. Of those who were blind, three were due to cataracts and one due to retinitis pigmentosa. While this sample is too small for statistical analysis, it may reasonably be expected that those who were cataract blind could be treated if provided with cataract surgery.

Refractive error was found in 66.4% of clients (*n* = 328) and was the cause of vision loss in 79% (*n* = 98) of cases. We hypothesise that the high prevalence of refractive error in the present study, when compared with population research, is attributable to selection bias plus limited access to glasses for many refugees prior to their arrival in Australia. Most clients with refractive error in our study (92.3%, *n* = 303) were provided with glasses. This identifies uncorrected or under-corrected vision as a significant issue among newly arrived refugees. The provision of glasses by the Essilor Vision Foundation through RHS has created outstanding access to products which many clients could not access due to cost and other service provision barriers. This improves the experience of resettlement in Australia and the ability to learn English [4,5].

Cataracts were found in 6.7% (*n* = 30) of all clients. Bhutanese and Nepalese refugees were 4.8 times more likely to have cataracts compared to all other ethnicities (adjusted OR 4.8; 95% CI: 2.0–11.4; *p* < 0.0001), which has not been reported in other studies on refugee eye health, to our knowledge. Twenty-two clients with cataracts were referred to specialist services for potential surgery; however, the outcomes from these referrals are not known.

A diagnosis of dry eyes was found in 38.3% of clients, which is higher than what was found in a comparable study in 2019, which reported dry eyes to be in 11% of participants [20]. Dry eye was more prevalent among females, which corresponds with the published literature on this condition [27,28]. There is limited evidence regarding ethnicity and dry eyes, although Asian heritage may increase the risk, which may be why dry eyes were more common for Burmese, Bhutanese, and Afghan refugees [27]. Dry eye is generally considered to increase with age [27,28,29], a finding which was not supported by this study. Data regarding comorbidities and medications that may increase risk of dry eyes were not collected. Dry eye has a significant impact on quality of life and is associated with depression [30,31,32], and the treatment recommended to our clients included lubricants and/or omega 3 supplements to purchase over the counter by the client.

Using accredited interpreters is essential to provide quality health services for refugees and culturally and linguistically diverse populations [33] (p. 72). Since the data collection for this study concluded, there has been a change in policy where Federal Government-funded interpreters can be accessed by private optometrists. However, there are no data on whether private optometrists are adequately aware of or utilising this service for refugee clients. Furthermore, funded access to interpreters for optometrists is not available in all Local Government Areas (LGA). For these reasons, and as further discussed below with regard to hospital settings, we recommend that state and federal health departments consider strategies for (1) increasing the awareness among optometrists of their access to free interpreting services; (2) an expansion of geographical access to interpreters beyond the current LGAs; and (3) an extension of funding for interpreters beyond October 2026 (the end of the current funding cycle to our knowledge) [34].

Over one in ten clients needed ophthalmic reviews for conditions potentially requiring close monitoring and medical treatment (e.g., glaucoma), laser or intravitreal treatments (e.g., diabetic retinopathy or age-related macular degeneration), or surgery (e.g., cataracts, dermatochalasis). Many of these conditions lead to vision loss, some of which are irreversible without early ophthalmic intervention. Clients were referred to public ophthalmology services in Adelaide, as private care was unaffordable—a common barrier refugees experience when accessing specialist health care in Australia [35]. Comprehensive and free-of-charge, public ophthalmology services in Adelaide have limited availability, with median waiting times for an outpatient appointment ranging from five months to two years [36]. If clients require surgery, the wait time is even longer, with a median of 71 days after their initial appointment and 10% waiting over 337 days [37]. In hospital settings, in-person and over-the-phone interpreters are government-funded for most refugees but are underused, with as few as 20% of patients with poor English proficiency having access to an interpreter [38]. Accordingly, the Royal Australian and New Zealand College of Ophthalmologists has advocated for strategies to overcome such barriers and ensure equitable access to ophthalmic care for refugees and asylum seekers [23]. Refugee-specific ophthalmic services have the potential to achieve this, but, to our knowledge, only one such service exists across Australia [39].

There were no cases of vitamin A deficiency or trachoma detected clinically, which is comparable with recent studies and may be due to the improvements of vitamin A provision and trachoma elimination programs in refugee camps [4,14,40]. Globally, trachoma prevalence is decreasing, and the highest burden remains in sub-Saharan Africa [41]. One 2016 study in a refugee camp in Malawi (*n* = 635) found no cases of trachoma (14), yet one study in Ethiopia in 2014 (*n* = 2571) found 547 (21.3%) [13]. The lack of trachoma in this study may be due to the small sample of those from sub-Saharan Africa (*n* = 69; 14%). The Australian guidelines for new arrival refugee health assessment do not recommend routine laboratory testing for vitamin A deficiency [8]. Those who had dry eyes were tested for vitamin A deficiency by the medical staff at RHS; however, the data were not analysed.

The data for this study are somewhat outdated, from 2017 to 2018, and the demographic of refugees arriving in Australia is dynamic and dependent on world events and government policy. Therefore, this research cannot be generalised to all people from refugee backgrounds. Only 494 out of the 1400 clients in the study period were assessed by the optometrist and so we do not have data on undiagnosed ophthalmic conditions that could be identified through universal screening. Further studies on eye health through universal screening is recommended.

The relatively high prevalence of vision loss found in our study is likely due to selection bias, as this was a clinic-based (rather than population-based) study. All clients in the study were referred to optometry due to clinical concern, which has produced a biased sample. Given the ICD-11 was only developed in 2018, previous research has different categories of vision impairment (‘moderate/severe’ and ‘blindness’) which makes comparison challenging.

## 5. Conclusions

Vision loss is common among newly arrived refugees in Australia. Most of this vision loss is avoidable, most commonly with glasses, highlighting the need for universal eye screening for both adults and children, and having accessible optometry services for people from refugee backgrounds. Universal screening with visual acuity testing could be performed as part of the new arrival health assessment. Having access to accredited interpreters during eye health screening and optometry and ophthalmology assessment is essential to provide quality and equitable care. In addition, interpreters are essential for consumer engagement, medication safety and compliance, and ensuring informed consent. Without optimum vision the challenges associated with resettlement are accentuated.

## Figures and Tables

**Table 1 ijerph-21-00869-t001:** Demographic characteristics of those attending optometry, RHS, 2017–2018 (*n* = 494).

Characteristics		*n*	(%)
Sex	Female	261	(52.8)
	Male	233	(47.2)
Age group	0–9	46	(9.3)
	10–19	110	(22.3)
	20–29	60	(12.1)
	30–39	77	(15.6)
	40–49	110	(22.3)
	>50	91	(18.4)
Ethnicity	Afghan	104	(21.2)
	African	69	(14.0)
	Bhutanese/Nepalese	119	(24.1)
	Burmese	73	(14.8)
	Middle Eastern	129	(26.1)

RHS: Refugee Health Service.

**Table 2 ijerph-21-00869-t002:** Prevalence of presenting vision loss.

	Category of Vision Loss						
	Mild VI	Moderate VI	Severe VI	Blind	Total	Prevalence (%) (95% CI)	χ^2^ Test (*p*)
Total (*n* = 460)	58	59	3	4	124	27.0 (23.1–31.2)	
Sex							
Female (*n* = 245)	31	32	2	3	68	27.8 (22.2–33.4)	χ^2^ = 1.05 *p* = 0.903
Male (*n* = 215)	27	27	1	1	56	26.0 (20.6–31.4)	
Age group, years							
0–9 (*n* = 38)	5	4	0	0	9	23.7 (13.0–39.2)	χ^2^ = 40.17 *p* = 0.005
10–19 (*n* = 106)	9	7	0	0	16	15.1 (9.5–23.1)	
20–29 (*n* = 55)	5	6	0	0	11	20.0 (11.6–32.3)	
30–39 (*n* = 74)	14	7	2	1	24	32.4 (22.9–43.7)	
40–49 (*n* = 101)	13	14	0	0	27	26.7 (19.1–36.1)	
≥50 (*n* = 86)	12	21	1	3	37	43.0 (33.1–53.6)	

VI: Vision Impairment, RHS: Refugee Health Service.

**Table 3 ijerph-21-00869-t003:** Top three optometry diagnoses by ethnicity, RHS, 2017–2018.

Diagnosis			Ethnicity			Total *n* (%)
	Afghan *n* (%)	Bhutanese/ Nepalese *n* (%)	Burmese *n* (%)	African *n* (%)	Middle Eastern *n* (%)	
Refractive error	71 (68.3)	74 (62.2)	46 (63.0)	40 (58.0)	97 (75.2)	328 (66.4)
Dry eyes	43 (41.3)	56 (47.1)	36 (49.3)	22 (31.9)	32 (24.8)	189 (38.3)
Cataract	3 (2.9)	18 (15.1)	2 (2.7)	2 (2.9)	5 (3.9)	30 (6.1)

RHS: Refugee Health Service.

**Table 4 ijerph-21-00869-t004:** Vision loss by ocular symptoms, RHS, 2017–2018.

Symptoms	Presenting Vision		Total	Prevalence (%) with Vision Loss
	Normal Vision	Vision Loss		
	*n*	*n*	*n*	
Dry/gritty eyes	10	7	17	41.2
Poor vision	39	23	62	37.1
Blurred vision	123	71	194	36.6
Itchy eyes	39	21	60	35
Photophobia	8	4	12	33.3
Headaches	46	13	59	22
Watery eyes	63	17	80	21.3
Eye pain	51	10	61	16.4
Red eyes	17	2	19	10.5
No Symptoms	85	13	98	13.3
Other/unknown	1	0	1	0

RHS: Refugee Health Service.

## Data Availability

Access to the de-identified study data and a data dictionary will be considered upon reasonable request to the corresponding author.

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
