# Peer review of "Describing the Eye Health of Newly Arrived Refugees in Adelaide, South Australia"

_ijerph, 2024, doi:10.3390/ijerph21070869_

Round 1

Reviewer 1 Report

Comments and Suggestions for Authors

Thank you for the opportunity to review the manuscript by Murton et al which describes the eye health of a retrospective cohort of newly arrived refugees in Adelaide, South Australia between 2017-18. My comments are aimed to improve the calibre of the paper, but also the translatability of messages to the wider health professional community.

1 The cohort were selective in nature (494/1400;  35%) that were screened by the RHS optometry pathway. Do you have a sense of undiagnosed ophthalmic burden in the majority of non-symptomatic patients? Has there been any data of standardised ophthalmic screening within the service (or elsewhere) to characterise the true burden? This would be important also for both adults and children who may have lived with impaired vision for many years, so not have a sense of visual loss. 

2 Page 7 (line 262) it states that there were no cases of vitamin A deficiency or trachoma detected clinically. Was this confirmed with biochemical or infective screening (e.g. onchocerciasis) as there are subclinical cases reported, particularly in Bhutan/Afghanistan and/or is this part of routine RHS screening protocols? Dry and/or itchy eyes are associated with both diagnoses so would be important to clarify and/or discuss as a potential weakness/recommendation for screening.

3 What age ranges are seen in the RHS? Do you have a paediatric optometrist and/or awareness of burden of visual impairment of children seen overall? National guidelines state that all children and adolescents need vision assessment, but may not have developmental skills/capacity to report impairment (hence routine screening of all is required). Do you also have a sense of school entry visual screening and are those children referred back into the RHS if issues are detected?

4 What percentage of humanitarian entrants are seen through the SA RHS? Does this screening pathway also review asylum-seekers? If so, what percentage of those screened were from AS backgrounds? 

5 What percentage of the 494 patients screened had visual difficulties or impairments identified on overseas health assessments? It would also be useful to know what percentage had underlying chronic disease (e.g. diabetes) or history of prematurity/genetic conditions (latter for the children)?

6 Vision is also essential for resettlement linkages such as driving!

7 Did the Optometrist/RHS have embedded free medication access within the eye health pathway? Cost of lubricant drops etc can be costly and what assistance is in place for those diagnosed with dry eyes (or other conditions) which require medications? I note that you have embedded referral pathway for visual aids which is admirable.

8 Provision of professional interpreters is essential for health care in patients with limited English proficiency. There are more recent studies that have documented insufficient use of professional interpreters in children (Brophy Williams et al Journal of Paediatrics and Child Heath) as well as the Australian Commission for Safety and Quality Standards (2021) specifically related to organisations providing care for migrants and refugees. Whilst interpreters are important for equitable health care, they are also essential for safety, medication compliance, informed consent and engaging with consumers.

9 How could this model be replicated elsewhere in the country? What would be the barriers or facilitators based on your results (for both adults and children)?

Reviewer 2 Report

Comments and Suggestions for Authors

This is a very significant and important study.  The authors are to be congratulated for this study which will be of great use to those in the refugee health field.

The paper is set out clearly in methodology and discussion. The challenges faced by refugees in settlement due to vision loss is well articulated.

The only issue to comment on was in the conclusion it states that having access to interpreters and optometry and ophthalmology assessment is essential.

Given the study was looking at results from a RHS that was providing health care on arrival to refugees and asylum seekers in SA would it not be good to recommend that it be something that is incorporated into on arrival health care?  That is restating the point made on page 5 at line 174

Also a query in relation to identification of issues - on pate 3 at line 76 - it appears that it requires in the first instance a 'self-report' of issues, although it also states that this may arise during screening by a GP.  So was it both? Perhaps could be clearer.

Would the authors recommend that tests be done regardless? Especially as state on page 5 at 171 that a substantial proportion had vision loss including some who did not report symptoms.
